# Unique Pro-Inflammatory Response of Macrophages during Apoptotic Cancer Cell Clearance

**DOI:** 10.3390/cells9020429

**Published:** 2020-02-12

**Authors:** Veronica Mendoza-Reinoso, Dah Youn Baek, Adrianne Kurutz, John R. Rubin, Amy J. Koh, Laurie K. McCauley, Hernan Roca

**Affiliations:** 1Department of Periodontics and Oral Medicine, University of Michigan School of Dentistry, Ann Arbor, MI 48109, USA; 2Department of Pathology, University of Michigan Medical School, Ann Arbor, MI 48109, USA

**Keywords:** bone marrow macrophages, peritoneal macrophages, prostate cancer, bone metastasis, tumor associated macrophages, efferocytosis

## Abstract

The clearance of apoptotic cells by macrophages (efferocytosis) is crucial to maintain normal tissue homeostasis; however, efferocytosis of cancer cells frequently results in inflammation and immunosuppression. Recently, we demonstrated that efferocytosis of apoptotic prostate cancer cells by bone marrow-derived macrophages induced a pro-inflammatory response that accelerated metastatic tumor growth in bone. To evaluate the microenvironmental impact of macrophages and their efferocytic function, we compared peritoneal macrophages (P-MΦ) versus bone marrow-derived macrophages (BM-MΦs) using an efferocytosis in vitro model. The capability to engulf apoptotic prostate cells was similar in BM-MΦs and P-MΦs. Ex vivo analysis of BM-MΦs showed an M2-like phenotype compared with a predominantly M1-like phenotype in P-MΦs. A distinct gene and protein expression profile of pro-inflammatory cytokines was found in BM-MΦs as compared with P-MΦs engulfing apoptotic prostate cancer cells. Importantly, the reprogramming of BM-MΦs toward an M1-like phenotype mitigated their inflammatory cytokine expression profile. In conclusion, BM-MΦs and P-MΦs are both capable of efferocytosing apoptotic prostate cancer cells; however, BM-MΦs exert increased inflammatory cytokine expression that is dependent upon the M2 polarization stage of macrophages. These findings suggest that bone marrow macrophage efferocytosis of apoptotic cancer cells maintains a unique pro-inflammatory microenvironment that may support a fertile niche for cancer growth. Finally, bone marrow macrophage reprogramming towards M1-type by interferon-γ (IFN-γ) induced a significant reduction in the efferocytosis-mediated pro-inflammatory signature.

## 1. Introduction

Macrophage efferocytosis of apoptotic cells is essential for human health as it maintains tissue homeostasis by preventing the harmful effects of apoptotic cell accumulation and necrosis [1,2,3]. Conversely, efferocytosis of apoptotic cells in the tumor microenvironment, a perpetual process during tumor growth, promotes a pro-inflammatory and immunosuppressive program [4,5,6]. Our group recently identified that bone marrow-derived macrophage engulfment of cancer cells resulted in the production of inflammatory cytokines to promote immunosuppression and tumor growth in bone [7]. However, it remains unclear how the pro-tumorigenic and immunosuppressive effects of macrophage efferocytosis in bone metastasis are linked to uniqueness of the bone microenvironment.

Myeloid cells in the bone marrow provide a fertile microenvironment for cancer cells to stimulate metastatic establishment. Millions of cells undergo apoptosis in the tumor microenvironment; thus, macrophage-mediated efferocytosis of cancer cells has a crucial role in pro-tumorigenic and immunosuppressive mechanisms that lead to tumor progression and metastases [4,5]. Tumor-associated efferocytic macrophages are typically M2 polarized pro-inflammatory macrophages that promote cell growth through the production of IL-6, IL-23, and TNFα. They also drive tumor development through immune suppressive factors such as TGFβ and IL-10 [8,9]. Moreover, increased expression of MerTK, Axl, and Tyro3 receptor kinases by efferocytic macrophages in the tumor microenvironment stimulates tumor metastasis and immune suppression through elevated secretion of immune suppressive cytokines [10,11].

Although macrophages exhibit a high degree of plasticity [12,13], they are classified into two simplified categories: M1 (classically activated) or M2 (alternatively activated) phenotypes. M1 macrophages express CD80 and CD86 to trigger inflammation and tissue damage, whereas M2 macrophages express the mannose receptor-1 (CD206) and macrophage scavenger receptors (CD204 and CD163) to promote tissue remodeling and fibrosis [14]. Interferon-γ (IFNγ) has the ability to switch immunosuppressive tumor associated macrophages (TAMs), M2-like polarized macrophages, into M1-like immunostimulatory cells in different types of cancers [15,16].

Peritoneal macrophages are commonly used to study efferocytic responses after being exposed to apoptotic cells; yet, solid tumors like prostate cancer preferentially locate to the bone marrow versus the peritoneum. Recently, we determined that bone marrow macrophage engulfment of cancer cells resulted in a distinct profile of cytokine production. Engulfment of cancer cells, but not normal cells, resulted in the production of C-X-C motif chemokine ligand 5 (CXCL5) and other pro-inflammatory cytokines, which led to immunosuppression and rebounding tumor growth [7]. The possibility that tissue resident macrophages at different sites encounter, engulf, or present differing post-engulfment profiles upon apoptotic tumor cell efferocytosis is unknown. The prospect of considering the fundamental process of efferocytosis as a molecular switch in programming or re-programming tumor associated macrophages bears promise [17]. Such information could provide valuable new insights into the intervention of tumor progression.

Using primary macrophages and an efferocytosis in vitro model, we investigated the differential gene expression of inflammatory cytokines in response to apoptotic cancer cell clearance by bone marrow-derived and peritoneal macrophages. We found that bone marrow-derived macrophages have a very unique role in the efferocytosis-induced inflammatory gene expression profile, which is likely critical in the acceleration of prostate cancer skeletal metastasis.

## 2. Materials and Methods

### 2.1. Cell Lines and Animals

Murine (Ras+Myc)-induced prostate cancer (RM1)cells were a gift from Timothy C. Thompson (Baylor College of Medicine, Houston, TX, USA) [18,19]. C57BL/6J mouse primary prostate epithelial cells (mPEC) were obtained from Cell Biologics (C57-6038). Both cell lines were certified mycoplasma-free (RM1 cells were characterized by IDEXX BioResearch; mPECs were obtained from Cell Biologics). All animal experiments were performed with approval from the University of Michigan Institutional Animal Care and Use Committee. Immunocompetent C57BL/6J mice were purchased from the Jackson Laboratory.

### 2.2. Macrophage and Apoptotic Cell Co-Culture

Bone marrow-derived macrophages (BM-MΦs) were isolated from wildtype 4–6 week old male C57BL/6J mice by flushing the femur and tibia with minimum essential medium eagle - alpha modification (αMEM) supplemented with L-glutamine, antibiotic-antimycotic 1× and 10% fetal bovine serum (FBS). From the same mice, peritoneal macrophages (P-MΦs) were isolated by flushing the peritoneal cavity with 5 mL of ice cold αMEM (L-glutamine, antibiotic-antimycotic 1×, 10% FBS). BM-MΦs and P-MΦs were cultured in αMEM (L-glutamine, antibiotic-antimycotic 1×, 10% FBS) in the presence of macrophage colony stimulating factor (M-CSF) (30 ng/mL, #315-02, Peprotech, Rocky Hill, NJ, USA). After three days in culture, macrophages were plated independently at 1.5 × 10^6^ cells/well in αMEM (L-glutamine, antibiotic-antimycotic 1×, 0.25% FBS) for co-culture experiments. RM1 and mPEC cells were exposed to UV light for 30 min to induce apoptosis. Apoptotic (a) cells (>90% trypan blue incorporation) were co-cultured with macrophages at a 1:1 ratio in αMEM (L-glutamine, 0.25% FBS) for 16–18 h. For flow cytometric analyses, apoptosis was induced in RM1 cells and the cells were labeled with CellTrace™ CFSE (ThermoFisher Scientific, C34554), and then co-cultured with BM-MΦs or P-MΦs for 16–18 h. Efferocytosis inhibition was performed by incubation of cultures at 4 °C over 6 h.

### 2.3. Flow Cytometry

BM- and P-MΦs alone and co-cultured with RM1(a) cells were collected and incubated in fluorescence-activated cell sorter (FACS) staining buffer (phosphate buffered saline-1×, 0.2% bovine serum albumin). Antibodies and/or matched isotype controls were added and incubated for 1 h at 4 °C. The following antibodies were used: allophycocyanin (APC)-F4/80 (CI:A3-1) (#ab105080, Abcam, Cambridge, MA, USA); fluorescein isothiocyanate (FITC)-CD86 (GL-1) (#105006, Biolegend, San Diego, CA, USA), FITC-CD206 (C068C2) (#141704, Biolegend, San Diego, CA, USA). Cells were analyzed by flow cytometry using a BD FACSAriaTM III (BD biosciences, San Jose, CA, USA) and Amnis^®^ ImageStream^®^XMk II (Luminex, Austin, TX, USA).

### 2.4. Quantitative Polymerase Chain Reaction (qPCR)

Total RNA was isolated from BM- and P-MΦs alone, and co-cultured with RM1(a) or mPEC(a) cells using the RNeasy^®^ Mini Kit (#74104, Qiagen, Hilden, Germany) following the manufacturer’s instructions. qPCR was performed using TaqMan gene expression master mix (#4369016, AppliedBiosystems, Foster City, CA, USA) and the corresponding TaqMan probes: *Cxcl1* (Mm04207460_m1), *Cxcl4* (Mm00451315_g1), *Cxcl5* (Mm00436451_g1), *IL-6* (Mm00446190_m1), *CD86* (Mm00444540_m1), *CD206* (Mm01329362_m1), and *18S* (Mm03928990_g1). Real time PCR was analyzed on ABI PRISM 7700 (Applied Biosystems, Foster City, CA, USA). Relative expression levels were calculated after normalization to 18S expression.

### 2.5. Macrophage Reprogramming

BM-MΦs were harvested and expanded as described above. On day four, macrophages were stimulated for 24 h with 60 ng/mL of interferon-γ (IFN-γ, , 315-05, Peprotech, Rocky Hill, NJ, USA) in αMEM (L-glutamine, antibiotic-antimycotic 1×, 10% FBS, M-CSF 30 ng/mL) to reprogram BM-MΦs towards the M1-type. Efferocytosis assays were then performed by adding RM1(a) cells and co-cultured 16–18 h as described.

### 2.6. ELISA

CXCL1 and CXCL5 were quantitatively measured using RayBio^®^ Mouse enzyme-linked immunosorbent assay (ELISA) assay systems (#ELM-KC and #ELM-LIX, RayBiotech, Inc., Peachtree Corners, GA, USA) using the conditioned media collected from BM- and P-MΦs alone and in co-cultured with RM1(a) or mPEC(a) cells, and BM-MΦs alone and in co-culture treated with IFN-γ- and vehicle.

### 2.7. Statistics

Statistical analyses were performed using GraphPad Prism 6 (GraphPad Software, version 8.0.2, San Diego, CA, USA) using one-way analysis of variance (ANOVA) with Dunnet’s multiple-comparisons and unpaired t-tests with significance of *p* < 0.05.

## 3. Results

### 3.1. Bone Marrow-Derived and Peritoneal Macrophages Display Effective Efferocytosis of Apoptotic Prostate Cancer and Normal Prostate Cells

Efferocytosis of apoptotic cells by bone marrow-derived macrophages (BM-MΦs) and peritoneal macrophages (P-MΦs) has been previously demonstrated by flow cytometry analysis [7,20,21,22]. The ability of P-MΦs versus BM-MΦs to efferocytose apoptotic cancer and normal prostate epithelial cells was analyzed using primary BM-MΦs, isolated from C57BL/6J mouse femurs and tibiae, and P-MΦs, isolated from peritoneal exudates, in co-culture with apoptotic RM1(a) prostate cancer cells and apoptotic normal prostate epithelial cells mPEC(a). In addition, efferocytosis of live RM1(l) cells by BM and P-MΦs was also analyzed and compared with apoptotic RM1(a) cells. RM1 cells were derived from the prostate epithelium of C57BL/6J mice and overexpress *Ras* and *Myc* oncogenes that resemble the oncogene-specific gene expression signatures of prostate cancer patient samples, and these are associated with prostate cancer progression [23,24]. RM1 cells have been used in vossicle and intratibial mouse models, where cancer cells are implanted directly in the bone niche to study the interaction between tumor and bone at the early stages of skeletal tumor development [7,25]. The mPEC cells are primary prostate epithelial cells derived from the prostate tissue of C57BL/6J mice (Cell Biologics). RM1 and mPEC cells were exposed to UV light to induce apoptosis, and then live RM1(l), apoptotic RM1(a), and apoptotic mPEC(a) cells were pre-labeled with CFSE dye and co-cultured with BM- and P-MΦs. After 16–18 h, the cells were collected; labeled with anti-F4/80-APC or its IgG isotype control; and analyzed using FACS (BD FACSAria™ III) and ImageStream flow cytometry (Amnis), which provides microscopic event images (model workflow, Figure 1A). Figure 1B,C depict the results from double-labeled APC+CFSE+ cells, indicating partial or complete engulfment of live RM1(l), apoptotic RM1(a) and mPEC(a) cells by BM- and P-MΦs. The double positive APC+CFSE+ (light blue cells in flow scatter plots) represent the RM1(l), RM1(a), and mPEC(a) cells (CFSE+) that are engulfed by F4/80-APC+ macrophages in the early (E-gate) and late (L-gate) internalization stages (Figure 1B). BM- and P-MΦs engulfed a significantly higher percentage of mPEC(a) cells, however, the efferocytosis efficiency was similar in P-MΦs and BM-MΦs. Engulfment of live RM1(l) cells by BM- and P-MΦs was observed, however, the percentages were significantly lower when compared with apoptotic RM1(a) cells. It is important to understand that dead cells are always present in live cell cultures, and that the CFSE labeling procedure may have also caused RM1 cell death. Also, BM-MΦs (BM+) and P-MΦs (P+) (red, APC+) internalization stages of apoptotic RM1(a) and mPEC(a) cells (green, CFSE+) were corroborated by ImageStream, as shown in Figure 3C (late stage, L; early stage, E).

### 3.2. Bone Marrow-Derived Macrophages Display a Robust M2 Polarization

A previous study suggested that BM-MΦs are predominantly M2 (alternatively activated), while P-MΦs are M1 (classically activated) macrophages [26]. The M2-like polarized macrophages have been characterized by their ability to release anti-inflammatory cytokines [27], and recent studies implicated efferocytosis of apoptotic tumor cells in accelerated tumor growth by inducing M2 macrophage polarization [5,21]. To characterize M1/M2 polarization phenotypes in BM-MΦs and P-MΦs, we compared the expression of the surface markers CD86 (M1-type) and CD206 (M2-type) by flow cytometry. Ex vivo total bone marrow cells and peritoneal exudates were stained with anti-F4/80-APC, anti-CD86-FITC, anti-CD206-FITC, and their respective IgG-isotype controls and analyzed by flow cytometry. We found that approximately 1% of total bone marrow-derived cells and 29% of peritoneal exudate cells were F4/80^+^ (phagocytic MΦs). BM-MΦs expressed low CD86 levels (<5%, F4/80+CD86+, Figure 2A) and high CD206 levels (>60%, F4/80+CD206+, Figure 2A); in contrast, P-MΦs expressed high CD86 levels (>55%, F4/80+CD86+, Figure 2A) and low CD206 levels (<9%, F4/80+CD206+, Figure 2A), confirming that BM-MΦs and P-MΦs are predominantly M2- and M1-like polarized macrophages, respectively. In addition, flow cytometric analysis of CD86 and CD206 expression was performed in BM-MΦs and P-MΦs cultured in vitro in the presence of M-CSF for three days and exposed to RM1(a) cells. In order to investigate how efferocytosis affects macrophage polarization, at day four, apoptotic RM1 cells were added to BM-MΦs and P-MΦs and cultured for 18 h. Interestingly, we found that approximately 50% of BM-MΦs and 70% of P-MΦs were F4/80+ (Figure 2B). In vitro, F4/80+ BM-MΦs maintained significantly higher CD206 expression levels (>80%, F4/80+CD206+, Figure 2B) when compared with P-MΦs (<14.5%, F4/80+CD206+, Figure 2B); however, P-MΦs cultured in vitro with M-CSF decreased CD86 expression levels (<7%, F4/80+CD86+, Figure 2B). Moreover, CD86 expression levels in BM-MΦ and P-MΦ co-cultured with apoptotic RM1 cells remained unchanged, although CD206 expression was slightly increased in P-MΦ co-cultured with apoptotic RM1 cells (>15%, F4/80+CD86+, Figure 2B). These results suggested that BM-MΦs maintained the M2-like phenotype, while P-MΦs showed neither M1- nor M2-like polarization in the in vitro cultures. Moreover, efferocytosis of apoptotic RM1 cells by P-MΦs slightly changed their polarization towards M2, whereas efferocytic BM-MΦs maintained their strong M2 polarization.

### 3.3. Bone Marrow Macrophage Efferocytosis of Apoptotic Prostate Cancer Cells Induces Unique Pro-Inflammatory Cytokine Gene Expression

We previously demonstrated that BM-MΦ efferocytosis of apoptotic prostate cancer cells orchestrates a pro-inflammatory response in prostate skeletal metastasis. In that study, cytokine array analyses of efferocytic bone marrow-derived macrophages identified the upregulation of pro-inflammatory cytokines such as CCL5, CXCL1, CXCL5, IL-6, and IL-12 [7]. We hypothesized that BM-MΦs efferocytosis of apoptotic cancer cells may induce a different pro-inflammatory cytokine response when compared with P-MΦs efferocytosis. To test this hypothesis, we assessed cytokine expression in BM- and P-MΦs in co-cultures with apoptotic prostate cancer RM1(a) or non-cancer prostatic epithelial mPEC(a) cells. Both BM- and P-MΦs were isolated from the same mice and cultured under the same conditions. Relative mRNA expression was analyzed by qPCR for Cxcl1, Cxcl4, Cxcl5, and IL-6, and macrophage responses to efferocytosis were compared. Figure 3A shows that Cxcl1, Cxcl4, Cxcl5, and IL-6 were significantly upregulated in co-cultures of BM-MΦs and RM1(a) cells in contrast to mPEC(a) cells. Conversely, Cxcl1 and Cxcl5 expression levels remained unaffected in P-MΦs co-cultured with RM1(a) or mPEC(a) cells. Cxcl4 and IL-6 expression was significantly increased in P-MΦs co-cultured with RM1(a) versus mPEC(a) cells, but was lower in comparison with BM-MΦs (Figure 3A). In addition, CXCL1 and CXCL5 were evaluated by ELISA in the conditioned media from BM- and P-MΦs co-cultured with apoptotic prostate cancer RM1(a) or non-cancer prostatic epithelial mPEC(a) cells (Figure 3B). Significantly higher levels of CXCL1 and CXCL5 were found in BM-MΦs co-cultured with RM1(a) cells; however, no changes were observed with mPEC(a) (Figure 3B). Interestingly, CXCL1 levels were increased in P-MΦs co-cultured with apoptotic RM1(a) or mPEC(a) cells relative to P-MΦs alone, although the fold stimulation was significantly lower as compared with BM-MΦs (Figure 3B). It has been shown that macrophage efferocytosis blockade of highly apoptotic cancer cells at 4 °C reduces the expression of M2-like macrophage associated genes [21]. To confirm that macrophage efferocytosis of apoptotic cancer cells drives the expression of pro-inflammatory cytokines, BM-MΦs and P-MΦs were incubated alone and co-cultured with RM1(a) cells at 37 °C and at 4 °C, where efferocytosis is inhibited. The gene expression of pro-inflammatory Cxcl1, Cxcl4, Cxcl5, and IL-6 was quantified by qPCR. Figure 3C shows that BM-MΦs efferocytosis of RM1(a) cells significantly increased Cxcl1, Cxcl5, and IL-6 gene expression under normal conditions (37 °C). Moreover, efferocytosis inhibition at 4 °C significantly decreased Cxcl5 expression levels in BM-MΦs. Basal gene expression levels of Cxcl1, Cxcl4, and IL6 in BM-MΦs alone increased at 4°C; however, no stimulation was observed with RM1(a) in Cxcl1 and IL-6 expression under inhibitory conditions. Interestingly, it has been demonstrated that pro-inflammatory cytokines CXCL1, CXCL4, CXCL5, and IL-6 trigger tumor progression in different contexts [7,28,29,30]. These results suggest that the higher expression of inflammatory cytokines triggered by efferocytosis of apoptotic cancer cells is uniquely associated with bone marrow-derived macrophages.

### 3.4. M1 Bone Marrow Macrophage Repolarization by INF-γ Reduces Pro-Inflammatory Cytokine Expression

Bone marrow macrophages have the capacity to switch phenotypes in response to interferon- γ (INF-γ) and/or lipopolysaccharides (LPS) [28,29]. We hypothesized that M1 repolarization of BM-MΦs (predominantly M2 polarized) would affect the inflammatory cytokine profile induced after efferocytosis of RM1(a) cells. To test this, BM-MΦs were isolated from long bones of C57BL/6J mice and cultured in the presence of M-CSF for 72 h. IFN-γ was then added and incubated for 24 h, at which point RM1(a) cells were added and co-cultured with BM-MΦs for 16–18 h (workflow, Figure 4A). To confirm the BM-MΦs’ reprogramming, we performed a flow cytometric analysis of BM-MΦs co-cultured with apoptotic cells in the presence of IFN-γ or vehicle control. Cells were labeled using F4/80-APC (MΦs) in combination with CD86-FITC (M1) or CD206-FITC (M2) and control IgG antibodies (Figure 4B). BM-MΦs treated with IFN-γ showed significantly higher CD86 (>35%, F4/80^+^CD86^+^, Figure 4B) and lower CD206 (~45%, F4/80^+^CD206^+^, Figure 4B) expression levels when compared with those vehicle treated (<12% F4/80^+^CD86^+^; >92% F4/80^+^CD206^+^; Figure 4B). This confirmed the BM-MΦs’ repolarization towards an M1 profile. BM-MΦs’ responses to efferocytosis of apoptotic cells in IFN-γ- and vehicle-treated cultures were analyzed by relative mRNA expression for CD86, Cxcl1, Cxcl4, Cxcl5, and IL-6 (Figure 4C). Figure 4C shows that IFN-γ treatment of BM-MΦs alone and BM-MΦs co-cultured with RM1(a) cells significantly increased CD86 expression, in correlation with the flow cytometric analysis (Figure 4B). With the exception of IL-6 (no change), gene expressions of the inflammatory cytokines (Cxcl1, Cxcl4, and Cxcl5) were significantly reduced in BM-MΦs alone and BM-MΦs co-cultured with RM1(a) cells when treated with IFN-γ (Figure 4C). Moreover, pro-inflammatory cytokine detection was performed by ELISA for CXCL1 and CXCL5 using the conditioned media from BM-MΦs alone and co-cultured with RM1(a) and IFN-γ- and vehicle-treated cultures. The results confirmed the gene expression data from Figure 4C. CXCL1 and CXCL5 expression was significantly decreased in BM-MΦs alone and BM-MΦs co-cultured with RM1(a) after IFN-γ treatment (Figure 4D). Altogether, these results suggest that BM-MΦ repolarization towards M1 profile mitigates the inflammatory response mediated by efferocytosis of apoptotic prostate cancer cells and suggest the dependence of this response on the M2-activation stage of macrophages.

## 4. Discussion

Recent investigations have indicated that bone marrow-derived macrophage efferocytosis of apoptotic cancer cells stimulates bone metastasis through pro-inflammatory and immunosuppressive responses [7,30]. It remains unclear how the bone marrow microenvironment shapes these metastatic responses and how bone marrow macrophages uniquely mediate the efferocytosis-accelerating tumor growth. An in vitro efferocytosis model was used to evaluate the differential responses of bone marrow-derived and peritoneal macrophage efferocytosis of apoptotic cancer cells and demonstrated that bone marrow-derived macrophages activate a unique pro-inflammatory response upon apoptotic cancer cell clearance.

Few studies have identified key differences between bone marrow-derived and peritoneal macrophage properties in different contexts. In 2013, a study demonstrated that bone marrow-derived macrophages are more proliferative and have higher phagocytic rates when compared with peritoneal and spleen macrophages [31]. However, in the current study, similar efferocytosis efficiencies of apoptotic prostate cancer cells were found in bone marrow-derived versus peritoneal macrophages. The ImageStream analysis identified early (high CFSE intensity) and late (low CFSE intensity) stages of cancer apoptotic cell engulfment by F4/80^+^ macrophages. Additionally, they identified that in vitro generated bone marrow-derived macrophages have higher expression of TGF-β and IL-10, intimating their M2-like phenotype [31]. These data were corroborated in 2016 by a study that demonstrated that bone marrow-derived macrophages display higher CD206 (M2-like marker) compared with peritoneal macrophages, which was also confirmed by lower oxidized low-density lipoprotein receptor (LOX-1, M1-like marker) and higher peroxisome proliferator-activated receptor-γ (PPARγ, M2-like marker) expression in the bone marrow-derived macrophages [26]. Similarly, we found that bone marrow-derived macrophages consistently expressed a higher percentage of F4/80^+^CD206^+^ (M2-like) in ex vivo and in vitro experiments, whereas peritoneal macrophages expressed a higher percentage of F4/80^+^CD86^+^ (M1-like) ex vivo, but with no clear phenotype in vitro. This may be explained by the presence of M-CSF in the media, which is known for its effect in promoting M2 polarization in macrophages [20]. Interestingly, another study demonstrated that bone marrow-derived macrophage efferocytosis of apoptotic colon cancer cells resulted in an increase of M2-like macrophages promoting tumor metastasis [32]. Efferocytosis performed by tumor-associated macrophages (TAMs) promotes their polarization toward an M2-like wound healing phenotype triggering immunosuppressive signals within the tumor microenvironment [4,21,33]. In contrast, we found that P-MΦs co-cultured with apoptotic RM1 cells showed a slight increment in the percentage of F4/80+CD206+, likely owing to their polarization towards M2 induced by efferocytosis. The strong CD206 expression (M2 polarization) of BM-MΦs remains unchanged, which may be because of their already high percentage of F4/80+CD206+ (~90%) under basal conditions.

Bone marrow-derived macrophage efferocytosis of apoptotic cancer cells, but not apoptotic normal cells, resulted in increased gene expression of *Cxcl1*, *Cxcl4*, *Cxcl5*, and *IL-6* inflammatory cytokines in comparison with peritoneal macrophages. In line with these results, ELISAs measuring CXCL1 and CXCL5 cytokines revealed elevated levels in bone marrow-derived macrophages upon apoptotic RM1 cancer cell engulfment. A significantly reduced stimulation was observed by efferocytic peritoneal macrophages, which could be explained by their polarization towards M2, as shown in Figure 2B. Moreover, efferocytosis inhibition of bone marrow-derived macrophages (4°C) resulted in decreased *Cxcl5* and unchanged *Cxcl1* and *IL-6* expression, which confirmed that the pro-inflammatory response of bone marrow-derived macrophages is produced by efferocytosis of apoptotic cancer cells. Upregulation of *Cxcl1* and *IL-6* gene expression in bone marrow-derived macrophages alone at 4°C may be the result of acute cold stress, which decreases IFN-γ expression [34]. These results correlated with recent in vitro and in vivo clinical data, where bone marrow-derived macrophage efferocytosis of cancer cells promotes the expression of CCL5, CXCL1, CXCL5, and IL-6 pro-inflammatory cytokines that accelerate tumor progression in the bone microenvironment [7]. The roles of these cytokines have been previously associated with tumor inflammation mechanisms and metastasis in various cancer types [7,35,36,37,38]. In accordance with this hypothesis, we found that inducing bone marrow macrophage repolarization with IFN-γ towards M1-type resulted in a significant downregulation of *Cxcl1*, *Cxcl4*, and *Cxcl5* expression in efferocytic macrophages, which was corroborated by the CXCL1 and CXCL5 ELISA results. M1 polarized macrophages have been described as anti-tumorigenic macrophages that remove tumor cells and produce high levels of immunostimulatory cytokines [13]. Macrophages treated with IFNγ, LPS, and IFNγ + LPS promote M1-like macrophage polarization [13,39]. Macrophage polarization (M1/M2) plasticity is ideal for immunomodulation of these cells in diseases involving macrophage dysregulation, such as cancer. Novel studies are reporting the use of macrophage-reprogramming agents to promote M1-like TAMs polarization that reduce tumor growth [40,41,42]. Similarly, the findings presented here suggest that efferocytic macrophages in the tumor bone microenvironment may be reprogrammed towards the M1 phenotype to reduce inflammation and tumor progression [17].

In summary, bone marrow macrophage efferocytosis of apoptotic prostate cancer cells drives a unique inflammatory response. This response as well as their polarization stage differs from the response of other site-specific macrophages such as peritoneal macrophages. As shown before, efferocytosis of apoptotic cancer cells accentuates M2 polarization of bone marrow macrophages [21], which accelerates tumor promoting inflammation and immunosuppression, suggesting that bone marrow macrophages may play a critical role during progression of skeletal metastasis. Bone marrow macrophage reprogramming towards M1 reduces the efferocytosis-mediated pro-inflammatory phenotype. We are currently pursuing studies to characterize the exact molecular mechanisms mediating bone marrow macrophage efferocytosis of apoptotic cancer cells in order to identify novel drug targets to be used as coadjuvant therapies to treat skeletal metastasis.

## Figures and Tables

**Figure 1 cells-09-00429-f001:**
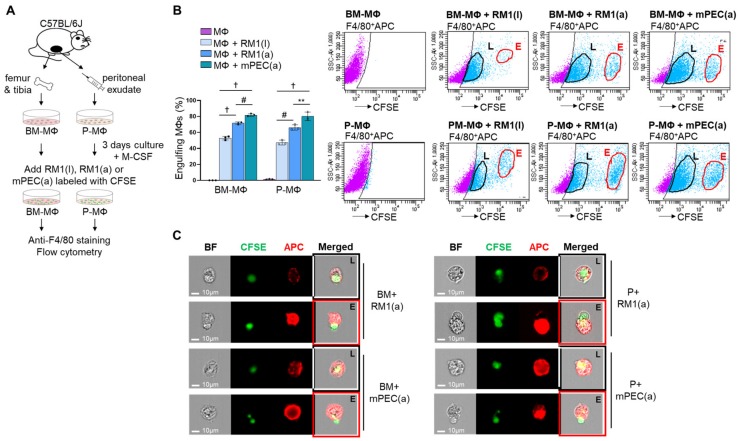
Efferocytosis of apoptotic prostate cancer cells and normal prostate epithelial cells by bone marrow-derived (BM) and peritoneal (P) macrophages (MΦ). (**A**) BM-MΦ and P-MΦ isolation and in vitro culture. Prostate cancer (RM1) and normal prostate ephitelial (mPEC) cells were exposed to UV to induce apoptosis and RM1(l), RM1(a), and mPEC(a) were stained with cell-trace-CFSE. BM-MΦ and P-MΦ were co-cultured with live prostate cancer RM1(l), apoptotic prostate cancer RM1(a) cells, and apoptotic normal prostate epithelial mPEC(a) cells. The co-cultures were stained with anti-F4/80-APC. (**B**) Flow cytometry analysis from A. Gates were adjusted to IgG isotype control. (**C**) Representative images from ImageStream show macrophages (F4/80-APC^+^, red) engulfing RM1(a) and mPEC(a) cells (CSFE^+^, green) in the early (E, red line) and late (L, black line) internalization stages. BF, bright field; BM+, bone marrow macrophages; P+, peritoneal macrophages. Data are mean ± SEM, *n* = 4 per group; ***p* < 0.01, #*p* < 0.001, †*p* < 0.0001 (one-way analysis of variance (ANOVA); Dunnet’s multiple-comparisons test).

**Figure 2 cells-09-00429-f002:**
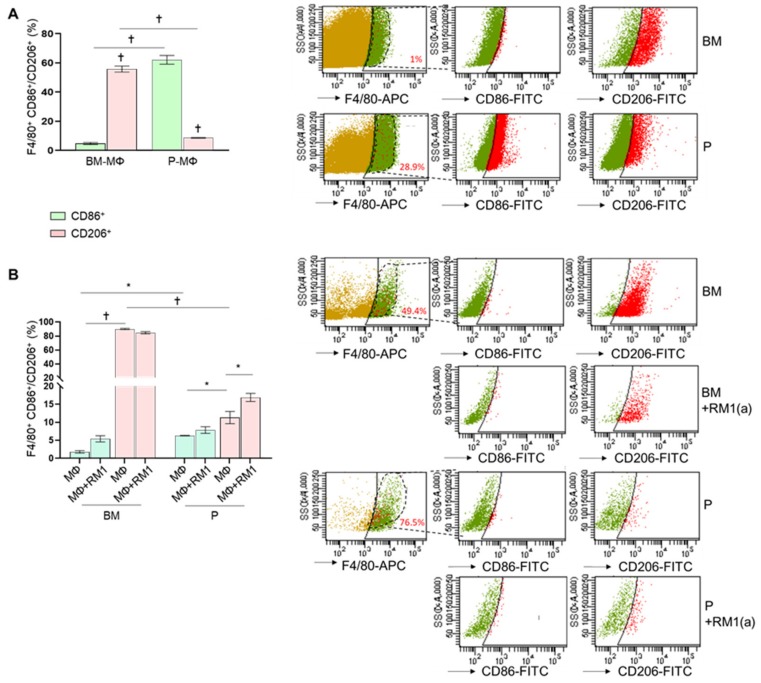
Ex vivo and in vitro M1 and M2 polarization of bone marrow-derived and peritoneal macrophages. BM-MΦs and P-MΦs were isolated from C57BL/6J mice. (**A**) Freshly isolated bone marrow and peritoneal exudate cells were stained for anti-F4/80-APC combined with either anti-CD86-FITC or anti-CD206-FITC and analyzed by flow cytometry. (**B**) BM-MΦs and P-MΦs co-cultured with RM1(a) cells for 18 h were stained with anti-F4/80-APC combined with either anti-CD86-FITC or anti-CD206-FITC and analyzed by flow cytometry. BM, bone marrow; P, peritoneal. Data in A and B are mean ± SEM, *n* = 3 per group; **p* < 0.05, †*p* < 0.0001 (one-way ANOVA; Dunnet’s multiple-comparisons test).

**Figure 3 cells-09-00429-f003:**
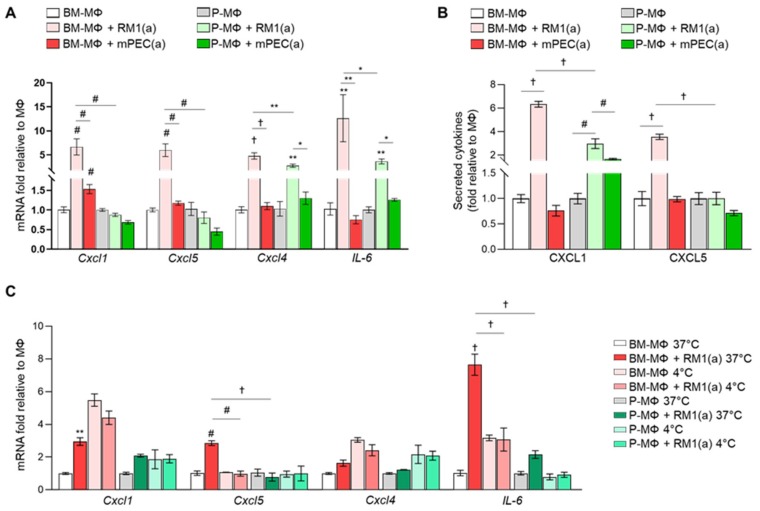
Differential gene expression of bone marrow-derived and peritoneal macrophages after efferocytosis of apoptotic prostate cancer cells and normal prostate epithelial cells. (**A**) mRNAs isolated from BM-MΦs and P-MΦs alone or co-cultured with apoptotic prostate cancer cells RM1(a) or apoptotic prostate epithelial cells mPEC(a) for 18–20 h were analyzed by quantitative PCR (qPCR) for the specified genes. (**B**) Conditioned media from (**A**) were analyzed by ELISA for total CXCL1 and CXCL5 levels. (**C**) qPCR analysis of BM-MΦs and P-MΦs during efferocytosis permissive (37 °C) or inhibitory (4 °C) conditions. Graphs show the gene expression quantification of inflammatory cytokines normalized to respective MΦs alone. Data are mean ± SEM, *n* = 3 per group; **p* < 0.05, ***p* < 0.01, #*p* < 0.001, †*p* < 0.0001 (one-way ANOVA; Dunnet’s multiple-comparisons test).

**Figure 4 cells-09-00429-f004:**
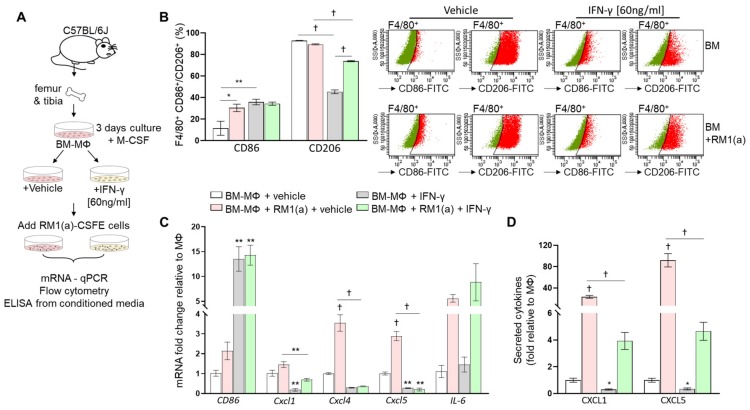
Effect of interferon-γ (IFNγ)-M1 polarization of bone marrow-derived macrophages in inflammatory cytokine expression. (**A**) Workflow: BM-MΦs were isolated and enriched in M-CSF for three days, and then treated with vehicle or IFN-γ and cultured alone or co-cultured with RM1(a) cells for 16–18 h. (**B**) BM-MΦs alone and in co-culture with RM1(a) cells were stained for anti-F4/80-APC combined with either anti-CD86-FITC or anti-CD206-FITC and analyzed by flow cytometry. (**C**) mRNAs from (**A**) were isolated and analyzed by qPCR for the specified inflammatory cytokine genes. (**D**) Conditioned media from (**A**) were analyzed via ELISA for total CXCL1 and CXCL5 levels. Data are mean ± SEM, *n* = 3 per group; **p* < 0.05, ***p* < 0.01, †*p* < 0.0001 (one-way ANOVA; Dunnet’s multiple-comparisons test and unpaired *t*-test).

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
