# Peer review of "Unique Pro-Inflammatory Response of Macrophages during Apoptotic Cancer Cell Clearance"

_cells, 2020, doi:10.3390/cells9020429_

Round 1

Reviewer 1 Report

In this study, the authors studied efferocytosis and M1/M2 polarization of bone marrow-derived and peritoneal macrophages and examined expression of several pro-inflammatory cytokines at RNA level by qPCR.  IFN-γ was used to reprogram the BM-MΦ to M1-type. The authors concluded “BM-MΦs and P-MΦs are both capable of efferocytosing apoptotic prostate cancer cells; however, BM-MΦs exert increased inflammatory cytokine gene expression that is dependent upon the M2 polarization stage of macrophages. These findings suggest a niche unique role of BM-MΦs in supporting metastatic growth where the bone microenvironment commands the macrophage phenotype creating a fertile environment for cancer growth. Macrophage reprogramming is presented as a candidate approach to overcome the efferocytosis-accelerating effects of metastatic tumor growth in bone.”

The manuscript is well written and organized. The results are clearly presented. However, some conclusions are overstated without substantial evidence.

Major issues:

1, “Unique Efferocytic Gene Expression Profiles of Bone Marrow-derived and Peritoneal Macrophages”: There are two concerns. One is RNA expression not necessary the same as protein expression, which should be validated by other means. The other issue is by examining several predetermined cytokines may miss other unique/differential expression factors/cytokines.

2, “These findings suggest a niche unique role of BM-MΦs in supporting metastatic growth where the bone microenvironment commands the macrophage phenotype creating a fertile environment for cancer growth. Macrophage reprogramming is presented as a candidate approach to overcome the efferocytosis-accelerating effects of metastatic tumor growth in bone.”

However, there is no experiment design to address those questions and no result from this manuscript supporting those conclusions. The M2 polarization has been reported in many types of cancers including prostate cancer, however, not all those tumors develop bone metastasis. The authors’ previous report (Ref. 7) did not fully address this issue. Intratibial injection of tumor is not the same as spontaneous metastasis of which tumor intrinsic factors play a critical role.  Besides M2-like MΦ, other tumor stromal factors /cells may contribute to distal metastasis as well. Those other possibilities should be at least discussed. 

Minor issues:

1, there are several typos or inconsistency of IFN-γ in the page 6 and Figure 3 legend.

Author Response

Response to Reviewer 1 Comments

We appreciate the thorough critique by reviewer one and the overall comments: “The manuscript is well written and organized. The results are clearly presented.”

We’ve addressed each comment and added data to substantiate conclusions.  As a result, we are confident that the manuscript is significantly improved and ready for dissemination to the greater community.

Please see changes in the revised manuscript attached.

Comment 1. “There are two concerns. One is RNA expression not necessary the same as protein expression, which should be validated by other means. The other issue is by examining several predetermined cytokines may miss other unique/differential expression factors/cytokines.”

Response 1. In response to the initial concern we have added new data of CXCL1 and CXCL5 cytokine levels evaluated by ELISA using conditioned media from apoptotic RM1 prostate cancer cells, BM (bone marrow)- and P (peritoneal)-macrophages (MΦs) alone and in co-culture with apoptotic RM1 cancer cells (see Figure 3 in the manuscript). We have also assessed CXCL1 and CXCL5 cytokine levels in conditioned media from BM-MΦs alone and in co-culture with apoptotic RM1 cells after IFN-γ and vehicle treatment (see Figure 4 in the manuscript). CXCL1 and CXCL5 are consistent with the gene expression analysis.

In response to the second concern, the main reason to analyze predetermined cytokines (CXCL1, CXCL4, CXCL5, and IL6) centers on the goal to demonstrate, for the first time, that there are significant differences in BM- and P-MΦ inflammatory responses upon efferocytosis of apoptotic RM1 cancer cells. Furthermore, and as outlined in the revised manuscript, these are critical cytokines for oncoimmune interactions. In studies beyond the scope of this one, we are working on single-cell sequencing experiments using resting and efferocytic BM-MΦs which is intended to identify unique genes and proteins involved in inflammation and tumor suppression upon efferocytosis.

Comment 2. “These findings suggest a niche unique role of BM-MΦs in supporting metastatic growth where the bone microenvironment commands the macrophage phenotype creating a fertile environment for cancer growth. Macrophage reprogramming is presented as a candidate approach to overcome the efferocytosis-accelerating effects of metastatic tumor growth in bone.”

“There is no experiment design to address those questions and no result from this manuscript supporting those conclusions.” This refers to the comment in the manuscript of a ‘unique niche role of BM-M in supporting metastatic growth’.  “M2 polarization has been reported in many types of cancers including prostate cancer, however, not all those tumors develop bone metastasis.” “Besides M2-like MΦ, other tumor stromal factors/cells may contribute to distal metastasis as well. Those other possibilities should be at least discussed.”

Response 2. We agree with this concern and have now revised the text to: “These findings suggest that bone marrow-derived macrophage efferocytosis of apoptotic cancer cells maintains a unique pro-inflammatory microenvironment that may support a fertile niche for cancer growth. Finally, bone marrow macrophage reprogramming towards M1-type by IFN-γ induced a significant reduction in the efferocytosis-mediated pro-inflammatory phenotype.” This can be found on page 1 (Abstract) of the revised manuscript.

Comment 3. There are several typos or inconsistency of IFN-γ in the page 6 and Figure 3 legend.

Response 3. IFNγ has been revised to IFN-γ throughout the manuscript.

Reviewer 2 Report

The paper by Mendoza-Reinoso V. et al., describes the different behaviour of bone marrow-derived macrophages and peritoneal derived-ones for what concerns the efferocytosis towards prostate cancer cells. They showed that both macrophage types are able to similarly engulf apoptotic prostate cells but BM  macrophages show increased expression of a pattern of inflammatory cytokines probably due to their M2 polarization. Despite the fact that the topic is interesting, several controls are missing. We suggest some additional experiments to improve the manuscript’s quality.

In fig 1B-C the authors show the engulfment of apoptotic RM1 cells by macrophages, indicating efferocytosis. In these experiments only tumor cells are used. We suggest to try the effect of macrophages on both non apoptotic tumor cells and on primary prostate epithelial cells, too.

In Fig 2 It is shown the M1 and M2 polarization of bone marrow-derived and peritoneal macrophages isolated from mice. We suggest to evaluate also the polarization in the co-culture condition since all the experiments are performed in the presence of tumor cells which most probably influence the macrophage polarization.

In Fig 4 only the CB86 marker is shown, additional markers for M1 polarization should be shown to prove that macrophages are really polarized towards the M1 phenotype.

The paper by Mendoza-Reinoso V. et al., describes the different behaviour of bone marrow-derived macrophages and peritoneal derived-ones for what concerns the efferocytosis towards prostate cancer cells. They showed that both macrophage types are able to similarly engulf apoptotic prostate cells but BM  macrophages show increased expression of a pattern of inflammatory cytokines probably due to their M2 polarization. Despite the fact that the topic is interesting, several controls are missing. We suggest some additional experiments to improve the manuscript’s quality.

In fig 1B-C the authors show the engulfment of apoptotic RM1 cells by macrophages, indicating efferocytosis. In these experiments only tumor cells are used. We suggest to try the effect of macrophages on both non apoptotic tumor cells and on primary prostate epithelial cells, too.

In Fig 2 It is shown the M1 and M2 polarization of bone marrow-derived and peritoneal macrophages isolated from mice. We suggest to evaluate also the polarization in the co-culture condition since all the experiments are performed in the presence of tumor cells which most probably influence the macrophage polarization.

In Fig 4 only the CB86 marker is shown, additional markers for M1 polarization should be shown to prove that macrophages are really polarized towards the M1 phenotype.

The paper by Mendoza-Reinoso V. et al., describes the different behaviour of bone marrow-derived macrophages and peritoneal derived-ones for what concerns the efferocytosis towards prostate cancer cells. They showed that both macrophage types are able to similarly engulf apoptotic prostate cells but BM  macrophages show increased expression of a pattern of inflammatory cytokines probably due to their M2 polarization. Despite the fact that the topic is interesting, several controls are missing. We suggest some additional experiments to improve the manuscript’s quality.

In fig 1B-C the authors show the engulfment of apoptotic RM1 cells by macrophages, indicating efferocytosis. In these experiments only tumor cells are used. We suggest to try the effect of macrophages on both non apoptotic tumor cells and on primary prostate epithelial cells, too.

In Fig 2 It is shown the M1 and M2 polarization of bone marrow-derived and peritoneal macrophages isolated from mice. We suggest to evaluate also the polarization in the co-culture condition since all the experiments are performed in the presence of tumor cells which most probably influence the macrophage polarization.

In Fig 4 only the CB86 marker is shown, additional markers for M1 polarization should be shown to prove that macrophages are really polarized towards the M1 phenotype.

The paper by Mendoza-Reinoso V. et al., describes the different behaviour of bone marrow-derived macrophages and peritoneal derived-ones for what concerns the efferocytosis towards prostate cancer cells. They showed that both macrophage types are able to similarly engulf apoptotic prostate cells but BM  macrophages show increased expression of a pattern of inflammatory cytokines probably due to their M2 polarization. Despite the fact that the topic is interesting, several controls are missing. We suggest some additional experiments to improve the manuscript’s quality.

In fig 1B-C the authors show the engulfment of apoptotic RM1 cells by macrophages, indicating efferocytosis. In these experiments only tumor cells are used. We suggest to try the effect of macrophages on both non apoptotic tumor cells and on primary prostate epithelial cells, too.

In Fig 2 It is shown the M1 and M2 polarization of bone marrow-derived and peritoneal macrophages isolated from mice. We suggest to evaluate also the polarization in the co-culture condition since all the experiments are performed in the presence of tumor cells which most probably influence the macrophage polarization.

In Fig 4 only the CB86 marker is shown, additional markers for M1 polarization should be shown to prove that macrophages are really polarized towards the M1 phenotype.

Author Response

Response to Reviewer 2 Comments

Reviewer 2 acknowledges the interest of the manuscript topic and suggests additional experiments to improve the quality.  We have addressed these comments by adding new experiments to support the study and improve the quality of presentation.

Please see changes in the revised manuscript attached.

Comment 1. “In fig 1B-C the authors show the engulfment of apoptotic RM1 cells by macrophages, indicating efferocytosis. In these experiments only tumor cells are used. We suggest to try the effect of macrophages on both non apoptotic tumor cells and on primary prostate epithelial cells, too.”

Response 1. In response to this comment we have performed an experiment including live prostate cancer cells (RM1) (yet with the understanding that there are always apoptotic cells in live cell cultures) and apoptotic normal prostate epithelial cancer cells (mPEC). Figure 1 now shows that BM- and P-MΦs engulfed significantly fewer ‘live’ RM1(l) cancer cells when compared to apoptotic RM1(a) and mPEC(a) cells (Figure 1B and C). Experimental processes before the detection of efferocytosis by flow cytometry may have provoked apoptosis of ‘live’ RM1 cells and hence why the engulfment is as positive as it is in the efferocytosis assay. The flow cytometric analysis of BM- and P-MΦs engulfing live and apoptotic prostate cancer (RM1(l), RM1(a)) and apoptotic normal prostate epithelial mPEC(a) cells are presented in Figure 1B and ImageStream figures of RM1(a) and mPEC(a) cells are presented in Figure 1C in the manuscript.

Comment 2. “In Fig 2 It is shown the M1 and M2 polarization of bone marrow-derived and peritoneal macrophages isolated from mice. We suggest to evaluate also the polarization in the co-culture condition since all the experiments are performed in the presence of tumor cells which most probably influence the macrophage polarization.”

Response 2. We agree this is a reasonable concern, and have included flow cytometric analysis graphs and statistics of CD86 and CD206 expression in BM- and P-MΦs alone and in co-culture with apoptotic RM1 cancer cells in Figure 2 (Page 5). Efferocytosis increased the polarization of P-MΦs towards M2, while strong M2 polarization in BM-MΦs remain unchanged. We have included the description of the results and graphs on page 4 and 5 and discussed the results on page 8.

Comment 3. “In Fig 4 only the CB86 marker is shown, additional markers for M1 polarization should be shown to prove that macrophages are really polarized towards the M1 phenotype.”

Response 3. We agree with this and we now include flow cytometric analysis data that show decreased CD206 (M2 marker) and increased CD86 (M1 marker) in BM-MΦs treated with IFN-γ (Figure 4B). This confirms the repolarization of BM-MΦs towards the tumor-suppressive M1 phenotype.

Reviewer 3 Report

The manuscript shows that BM-derived macrophages (BM-MΦs) and peritoneal macrophages (P-MΦs) can clear apoptotic prostate cancer cells through efferocytosis. Most BM-MΦs are M2 polarized. BM-MΦs and P-MΦs release different pro-inflammatory cytokines (CXCL1 and CXCL4) when co-cultured with RM1 prostate cancer cells. The different expression pattern is related to the polarization of macrophages. Overall the findings are interesting. The major defect is there is no evidence to show the varied pro-inflammatory cytokine expression of BM-MΦs is a result of efferocytosis. The detection must be repeated with a phagocytosis (efferocytosis) inhibitor.

There are mistakes in the description about Fig.2 in section 3.2. More specifically, the CD206 level of P-MΦs was actually high (line 159, page 4) according to the scatter image of flow cytometry (Fig.2A, the right image of low panel). It is "RM1 cells" but not "M-CSF" in line 162, page 4. Similarly, it is apoptotic prostate epithelial cells (mPECs) but not "cancer cells" in the legend of Fig. 3 (line 2000, page 6).

According to Fig. 1 and 4, the primary macrophages were cultured with 3d culture method. There is no description about 3d culture in section 2.2.

"These findings suggest a niche unique role of BM-MΦs in supporting metastatic growth where the bone microenvironment commands the macrophage
phenotype creating a fertile environment for cancer growth. Macrophage reprogramming is presented as a candidate approach to overcome the efferocytosis-accelerating effects of metastatic tumor growth in bone" ( Abstract, line 25 - 29, page 1). The last sentence is a premature conclusion and should be removed.

Author Response

Response to Reviewer 3 Comments

We’ve addressed each comment and added data to substantiate conclusions.  As a result, we are confident that the manuscript is significantly improved and ready for dissemination to the greater community.

Please find the revised manuscript attached including the modifications made.

Comment 1. “The major defect is there is no evidence to show the varied pro-inflammatory cytokine expression of BM-MΦs is a result of efferocytosis. The detection must be repeated with a phagocytosis (efferocytosis) inhibitor.”

Response 1. We addressed this important concern in additional experiments with an efferocytosis inhibition strategy where BM-MΦs and P-MΦs alone and in co-culture with apoptotic RM1 cells were incubated at 4°C to inhibit apoptotic cell engulfment by macrophages [1]. The results showed that gene expression of pro-inflammatory Cxcl1, Cxcl4, Cxcl5 and IL-6 was quantified by qPCR. Figure 3C shows that BM-MΦs efferocytosis of RM1(a) cells significantly increased Cxcl1, Cxcl5 and IL-6 gene expression under normal conditions (37°C). Moreover, efferocytosis inhibition at 4°C significantly decreased Cxcl5 expression levels in BM-MΦs. Basal gene expression levels of Cxcl1, Cxcl4 and IL6 in BM-MΦs alone increased at 4°C; however, no stimulation was observed with RM1(a) in Cxcl1 and IL-6 expression under inhibitory conditions. This has been added to the manuscript in section 3.3, Figure 3 description and discussion.

Comment 2. “There are mistakes in the description about Fig.2 in section 3.2. More specifically, the CD206 level of P-MΦs was actually high (line 159, page 4) according to the scatter image of flow cytometry (Fig.2A, the right image of low panel). It is "RM1 cells" but not "M-CSF" in line 162, page 4. Similarly, it is apoptotic prostate epithelial cells (mPECs) but not "cancer cells" in the legend of Fig. 3 (line 2000, page 6).”

Response 2. About the CD206 levels referred in line 159, page 4 (Figure 2A), these are shown as F4/80+CD206+P-MΦs (red dots on the bottom left scatter image, Figure 2A). The representative scatter plot may lead to the assumption that the amount of these cells (CD206+) is high in P-MΦs; however, CD206+ cells only represent approximately 9% (bar graph, Figure 2A) of the total F4/80+ P-MΦs population.

About line 162 page 4, thank you for noticing this, we have changed line 162 (now lines 177-179) to: “In addition, flow cytometric analysis of CD86 and CD206 expression was performed in BM-MΦs and P-MΦs cultured in vitro in the presence of M-CSF for 3 days and then stimulated with RM1(a) cells.” Also, Figure 3 legend is now changed to: “apoptotic prostate epithelial cells mPEC(a)”.

Comment 3. “According to Fig. 1 and 4, the primary macrophages were cultured with 3d culture method. There is no description about 3d culture in section 2.2.”

Response 3. We apologize for this lack of clarity. The Figures 1, 4 refer to ‘3d’ which signifies 3 days and not 3 dimensional (3D). This has been clarified in Figures 1 and 4 and is now indicated as ‘3 days culture’.     

Comment 4. "These findings suggest a niche unique role of BM-MΦs in supporting metastatic growth where the bone microenvironment commands the macrophage phenotype creating a fertile environment for cancer growth. Macrophage reprogramming is presented as a candidate approach to overcome the efferocytosis-accelerating effects of metastatic tumor growth in bone" (Abstract, line 25 - 29, page 1). The last sentence is a premature conclusion and should be removed.

Response 4. We agree with this comment and have revised the sentence to: “These findings suggest that bone marrow-derived macrophage efferocytosis of apoptotic cancer cells maintains a unique pro-inflammatory microenvironment that may support a fertile niche that allows cancer growth. Bone marrow-derived macrophage reprogramming reduces the efferocytosis-dependent pro-inflammatory phenotype.” on page 1 (Abstract, lanes 25-28).

Reviewer 4 Report

Mendoza-Reinoso and colleagues (Cells-657100) report on a in vitro cell biological study comparing efferocytosis (clearance of apoptotic cells) by bone-marrow versus peritoneal M-CSF1-derived macrophages.  The study follows a recent previous paper from these authors (Roca et al, JCI 2018) showing that bone-marrow derived macrophages (BMDM) can engulf dying prostate cancer cells, and in doing so, induce CXCL5 and other pro-inflammatory cytokines facilitate prostate tumor growth and metastasis.  In the present study, authors extend this study to compare (i) efferocytosis rates, (ii) M1/M2 markers, and (iii) a subset of cytokines induced following efferocytosis that include Cxcl1, Cxcl5, Cxcl4, and IL-6 in BMDM versus peritoneal macrophages (PM).  The major conclusions in the paper suggest that BMDM and PM both effectively engulf apoptotic tumor RM1 tumor cells, but they have different phenotypic properties and produce different cytokines.  Further studies suggest that both types of macrophages can differentially sense tumor apoptotic cells versus non-transformed apoptotic cells, and have different response to IFNg.

Overall, this is technically well performed, albeit somewhat preliminary and descriptive, that macrophages and efferocytosis is both dynamic and contextual.  While the study is interesting, and will stimulate research in this field, the paper presently may be a bit too preliminary to support the main conclusions.  A few issues came up in the review.

Authors must provide some assurances that cell lines (RM1 and mPECs) are mycoplasma free. Otherwise, pro-inflammatory cytokines are subject to mis-interpretation.

The studies comparing cytokines with apoptotic tumor cells versus apoptotic mPECs is also interesting (assuming mycoplasma free), although focus on a single cell line (RM1), and single method for induction of cell death (UV), limits the study.

Authors should better explain why only 4 cytokines was used, rather a luminex or multiplex that would provide much more information at the protein level.

Author Response

Response to Reviewer 4 Comments

We appreciate the comment that “overall, this is technically well performed” and “study is interesting and will stimulate research in this field”.  We also appreciate the concerns with a somewhat preliminary and descriptive nature of the work.  We hope the addition of new data and respectful consideration of changes in presentation allay these concerns and elevate the priority of this work.

Please find attached the revised manuscript including the changes made.

Comment 1. “Authors must provide some assurances that cell lines (RM1 and mPECs) are mycoplasma free. Otherwise, pro-inflammatory cytokines are subject to mis-interpretation.”

Response 1. In this study, RM1 cells were characterized by an independent laboratory (IDEXX BioResearch) which reported that these cells were mycoplasma free. mPECs were obtained from Cell Biologics (C57-6038), the data sheet of this cell line states: “Cells are negative for bacteria, yeast, fungi, and mycoplasma”. We have presented this information on page 2 (Material and methods).

Comment 2. “The studies comparing cytokines with apoptotic tumor cells versus apoptotic mPECs is also interesting (assuming mycoplasma free), although focus on a single cell line (RM1), and single method for induction of cell death (UV), limits the study.”

Response 2. In the pro-inflammatory cytokine gene expression analysis performed in efferocytic BM-MΦs and P-MΦs we used apoptotic RM1 and the results correlated with a previous study published by our group [2]. This study also used human prostate cancer (PC3) cells and identified similar changes in the pro-inflammatory cytokines gene expression. Since we were using primary macrophages from C57BL/6J mice for the in vitro experiments, we elected to use RM1 cells only in order to maintain the in vitro model using macrophages and cancer cells from the same species and genetic background. Moreover, RM1 cells overexpress Ras and Myc oncogenes that resembles the oncogene-specific gene expression signatures of prostate cancer patient samples, and these are associated with prostate cancer progression [3, 4]. Furthermore, RM1 cells have been used in vossicle models, where cancer cells are implanted directly in the bone niche to study the interaction between tumor and bone at early stages of skeletal tumor development [2, 5]. This is now added to the section 3.1 in the manuscript.

Similarly, our previous publication [2] induced RM1 apoptosis by UV treatment and by an apoptosis-inducible caspase 9 RM1 cells (RM1-iC9) method, where AP20187 (B/B Homodimerizer) treatment increased processed caspase-9 and cleaved caspase-3, which reflects apoptosis. The pro-inflammatory cytokine gene expression analysis using the alternative apoptosis-inducing method correlated with the results obtained from RM1 cells induced to apoptosis with UV light (Figure 2.D, [2]). Thus, we used the RM1 UV induction in order to avoid any interference of AP20187 in macrophages’ signaling.

Comment 3. “Authors should better explain why only 4 cytokines was used, rather a luminex or multiplex that would provide much more information at the protein level.”

Response 3. We previously demonstrated that BM-MΦ efferocytosis of apoptotic prostate cancer cells orchestrates a pro-inflammatory response in prostate skeletal metastasis. In the study, a cytokine array (40 target cytokines) analyses of efferocytic bone marrow-derived macrophages identified the upregulation of pro-inflammatory cytokines such us CCL5, CXCL1, CXCL5, IL-6, and IL-12 [2]. In the present study, we only analysed CXCL1, CXCL4, CXCL5, and IL6 cytokines based on the previous results and the focus for the work was to demonstrate, for the first time, that there are significant differences between BM- and P-MΦs in pro-inflammatory processes upon efferocytosis of apoptotic RM1 cancer cells. Although beyond the scope of this current study, future studies will indeed identify a more complex panel of mediators identified via single-cell sequencing experiments.

References

Soki, F.N., et al., Polarization of prostate cancer-associated macrophages is induced by milk fat globule-EGF factor 8 (MFG-E8)-mediated efferocytosis. J Biol Chem, 2014. 289(35): p. 24560-72. DOI: 10.1074/jbc.M114.571620. Roca, H., et al., Apoptosis-induced CXCL5 accelerates inflammation and growth of prostate tumor metastases in bone. J Clin Invest, 2018. 128(1): p. 248-266. DOI: 10.1172/JCI92466. Ju, X., et al., Novel oncogene-induced metastatic prostate cancer cell lines define human prostate cancer progression signatures. Cancer Res, 2013. 73(2): p. 978-89. DOI: 10.1158/0008-5472.CAN-12-2133. Koh, C.M., et al., MYC and Prostate Cancer. Genes Cancer, 2010. 1(6): p. 617-28. DOI: 10.1177/1947601910379132. Jung, Y., et al., Prevalence of prostate cancer metastases after intravenous inoculation provides clues into the molecular basis of dormancy in the bone marrow microenvironment. Neoplasia, 2012. 14(5): p. 429-39. DOI: 10.1596/neo.111740.

Round 2

Reviewer 2 Report

Authors have improved the quality of the manuscript and now it is ready for publication

Reviewer 3 Report

The manuscript has been vigorously revised. The new title is much better for the data presented. 

Reviewer 4 Report

The authors have addressed my main concerns.  The idea that different macrophage subsets produce differeny cyctokines during efferocytosis is now better supported and the paper will be interesting to this research field.